# Application of Transfer Learning in EEG Decoding Based on Brain-Computer Interfaces: A Review

**DOI:** 10.3390/s20216321

**Published:** 2020-11-05

**Authors:** Kai Zhang, Guanghua Xu, Xiaowei Zheng, Huanzhong Li, Sicong Zhang, Yunhui Yu, Renghao Liang

**Affiliations:** 1School of Mechanical Engineering, Xi’an Jiaotong University, Xi’an 710049, China; zhangkai0912@stu.xjtu.edu.cn (K.Z.); hlydx1314@stu.xjtu.edu.cn (X.Z.); lihuanzhong@stu.xjtu.edu.cn (H.L.); zhsicong@mail.xjtu.edu.cn (S.Z.); yuyunhui@stu.xjtu.edu.cn (Y.Y.); lrh8131@stu.xjtu.edu.cn (R.L.); 2State Key Laboratory for Manufacturing Systems Engineering, Xi’an Jiaotong University, Xi’an 710049, China

**Keywords:** EEG, transfer learning, review, decoding, classification

## Abstract

The algorithms of electroencephalography (EEG) decoding are mainly based on machine learning in current research. One of the main assumptions of machine learning is that training and test data belong to the same feature space and are subject to the same probability distribution. However, this may be violated in EEG processing. Variations across sessions/subjects result in a deviation of the feature distribution of EEG signals in the same task, which reduces the accuracy of the decoding model for mental tasks. Recently, transfer learning (TL) has shown great potential in processing EEG signals across sessions/subjects. In this work, we reviewed 80 related published studies from 2010 to 2020 about TL application for EEG decoding. Herein, we report what kind of TL methods have been used (e.g., instance knowledge, feature representation knowledge, and model parameter knowledge), describe which types of EEG paradigms have been analyzed, and summarize the datasets that have been used to evaluate performance. Moreover, we discuss the state-of-the-art and future development of TL for EEG decoding. The results show that TL can significantly improve the performance of decoding models across subjects/sessions and can reduce the calibration time of brain–computer interface (BCI) systems. This review summarizes the current practical suggestions and performance outcomes in the hope that it will provide guidance and help for EEG research in the future.

## 1. Introduction

A brain–computer interface (BCI) is a communication method between a user and a computer that does not rely on the normal neural pathways of the brain and muscles [1]. According to the methods of electroencephalography (EEG) signal collection, BCIs can be divided into three types, namely, non-invasive, invasive, and partially-invasive BCIs. Among them, non-invasive BCIs realize the control of external equipment via EEG and by transforming EEG recordings into a command, which have been widely used due to their convenient operation. Figure 1 shows a typical non-invasive BCI system framework based on EEG, which usually consists of three parts: EEG signal acquisition, signal decoding, and external device control. During this process, signal decoding is the key step to ensure the operation of the whole system.

The representation of EEG typically takes the form of a high-dimensional matrix, which includes the information of sampling points, channels, trials, and subjects [2]. Meanwhile, the most common features of EEG-based BCIs include spatial filtering, band power, time points, and so on. Recently, machine learning (ML) has shown its powerful ability for feature extraction in EEG-based BCI tasks [3,4].

BCI technology based on EEG has made great progress, but the challenges of weak robustness and low accuracy greatly hinder the application of BCIs in practice [5]. From the perspective of signal decoding, the reasons are as follows: First, one of the main assumptions of ML is that training and test data belong to the same feature space and are subject to the same probability distribution. However, this assumption is often violated in the field of bioelectric signal processing, because differences in physiological structure and psychological states may cause obvious variation in EEG. Therefore, signals from different sessions/subjects on the same task show different features and distribution.

Second, EEG signals are extremely weak and are always accompanied by unrelated artifacts from other areas of the brain, which potentially mislead discriminant results and decrease the classification accuracy. Third, the strict requirements for the experimental conditions of BCI systems make it difficult to obtain large and high-quality datasets in practice. It is difficult for a classification model based on small-scale samples to obtain strong robustness and high classification accuracy. However, large-scale and high-quality datasets are the basis for guaranteeing the decoding accuracy of models.

One promising approach to solve these problems is transfer learning (TL). The principle of TL is realizing the knowledge transfer from different but related tasks, i.e., using existing knowledge learned from accomplished tasks to help with new tasks. The definition of TL is as follows: A given domain *D* consists of a feature space *X* and a marginal probability distribution *P*(*X*). A task *T* consists of a label space *y* and a prediction function f. A source domain $D_{s}$ and a target domain $D_{T}$ may have different feature spaces or different marginal probability distributions, i.e., $X_{s}\neq X_{T}$ or $P_{s}\left(X\right)\neq P_{T}\left(X\right)$. Meanwhile, tasks $T_{s}$ and $T_{T}$ are subject to different label spaces. The aim of TL is to help improve the learning ability of the target predictive function $f_{T}$(·) in $D_{T}$ using the knowledge in $D_{s}$ and $T_{s}$ [6].

There are two main scenarios in EEG-based BCIs, namely, cross-subject transfer and cross-session transfer. The goal of TL is to find the similarity between new and original tasks and then to realize the discriminative and stationary information transfer across domains [7]. In this study, we attempted to summarize the transferred knowledge for EEG based on following three types: Knowledge of instance, knowledge of feature representation, and knowledge of model parameters.

This review of TL applications for EEG classification attempted to address the following critical questions: What problems does TL solve for EEG decoding? (Section 3.1); which paradigms of EEG are used for TL analysis? (Section 3.2); what kind of datasets can we refer to in order to verify the performance of these methods? (Section 3.3); what types of TL frameworks are available? (Section 3.4).

First, the search methods for the identification of studies are introduced in Section 2. Then, the principle and classification criteria of TL are analyzed in Section 3. Next, the TL algorithms for EEG from 2010 to 2020 are described in Section 4. Finally, the current challenges of TL in EEG decoding are discussed in Section 5.

## 2. Methodology

A wide literature search from 2010 to 2020 was conducted, resorting to the main databases, such as Web of Science, PubMed, and IEEE Xplore. The keywords used for the electronic search were TL, electroencephalogram, brain–computer interface, inter-subject, and covariate shift. Table 1 lists the collection criteria for inclusion or exclusion.

The search method of this review is shown in Figure 2, which was used to identify and to narrow down the collection of TL-based studies, resulting in a total of 246 papers. Duplicates between all datasets and studies without full-text links were excluded. Finally, 80 papers that meet the inclusion criteria were included.

## 3. Results

### 3.1. What Problems Does Transfer Learning Solve?

This review of the literature on TL applications for EEG attempted to address the following critical questions:

#### 3.1.1. The Problem of Differences across Subjects/Sessions

Although advanced methods such as machine learning have been proven to be a critical tool in EEG processing or analysis, they still suffer from some limitations that hinder their wide application in practice. Consistency of the feature space and probability distribution of training and test data is an important prior condition of machine learning. However, in the field of biomedical engineering, such as EEG based on BCIs, this hypothesis is often violated. Obvious variation in feature distribution typically occurs in representations of EEG across sessions/subjects. This phenomenon results in a scattered distribution of EEG signal features, an increase in the difficulty of feature extraction, and a reduction in the performance of the classifier.

#### 3.1.2. The Problem of Small Sample Size

In recent years, machine learning and deep neural networks have provided good results for the classification of linguistic features, images, sounds, and natural texts. A main reason for its success is that their massive amount of data guarantees the performance of the classifier. However, in practical applications of BCI, it is difficult to collect high-quality and large EEG datasets due to the limitations of strict requirements for the experimental environment and available subjects. The performance of these methods is highly sensitive to the number of samples; a small sample size tends to lead to overfitting during model training, which adversely affects the classification accuracy [8].

#### 3.1.3. The Problem of Time-Consuming Calibration

A large amount of data are required to calibrate a BCI system when a subject performs a specific EEG task. This requirement commonly takes a long calibration session, which is inevitable for a new user. For example, when a subject performs a steady-state visually evoked potential (SSVEP) speller task, the various commands cause a long calibration time. However, collecting calibration data is time-consuming and laborious, which reduces the efficiency of the BCI system.

### 3.2. EEG Paradigms for Transfer Learning

There are four paradigms of EEG-BCIs discussed in this paper and the percentage of these paradigms across collected studies are shown in Figure 3.

#### 3.2.1. Motor Imagery

Motor imagery (MI) is a mental process that imitates motor intention without real motion output [9], which activates the neural potential in primary sensorimotor areas. Different imagery tasks will induce potential activity in different regions of the brain. Thus, this response can be converted into a classification task. The feature of MI signals is often expressed in the form of frequency or band energy [10]. Due to task objectives and various feature representations, a variety of machine learning algorithms (e.g., deep learning and Riemannian geometry) can be applied to the decoding of MI [11,12].

#### 3.2.2. Steady-State Visually Evoked Potentials

When a human receives a fixed frequency of flashing visual stimuli, the potential activity of the cerebral cortex is modulated to produce a continuous response related to the frequency (same or multiples) of these stimuli. This physiological phenomenon is referred to a SSVEP [13]. Due to their stable and obvious representation of signals, BCI systems based on SSVEP are widely used to control equipment such as mobile devices, wheelchairs, and spellers.

#### 3.2.3. Event-Related Potentials

Event-related potentials are responses for multiple or diverse stimuli corresponding to specific meanings [14]. P300 is the most representative type of ERP, which occurs about 300 ms after a visual or auditory stimulus. A feature classification model can be used for decoding P300.

#### 3.2.4. Passive BCIs

A passive BCI is a form of interaction that does not rely on external stimuli. It achieves a brain control task by encoding the mental activity from different states of the brain [15]. Common types of passive BCI tasks include driver drowsiness, emotion recognition, mental workload assessment, and epileptic detection [16], which can be decoded by regression and classification models [17,18].

### 3.3. Case Studies on a Shared Dataset

Analysis between different datasets is not valid because they use different equipment or communication protocols. In addition, different mental tasks and collecting procedures also bring great differences to EEG. Therefore, the reviewed studies mainly concentrate on the TL across subjects/sessions in the same dataset. In Table 2, we briefly summarize the publicly available EEG dataset in this review.

### 3.4. Transfer Learning Architecture

In this review, we summarize previous studies according to “what knowledge should be transferred in EEG processing.” Multi-step processing for EEG across subjects/sessions results in discriminative information in different steps. Therefore, determining what should be transferred is the key problem according to different EEG tasks. Pan et al. [6] proposed authoritative classification approaches based on “what to transfer.” All papers collected in this review were classified according to this method (Figure 4). In the following sections, we have selected several representative methods for analysis.

#### 3.4.1. Transfer Learning Based on Instance Knowledge

It is often assumed that we can easily obtain large amounts of markup data from a source domain, but this data cannot be directly reused. Instance transfer approaches re-weight some source domain data as a supplement for the target domain. Based on instance transfer, the majority of the literature utilized the measurement method to evaluate the similarity between data from the source and target domains. The similarity metric was then converted into the transfer weight coefficient, which was directly used to instance transfer by re-weighting the source domain data [30,31,32]. Herein, we have listed a few typical methods based on instance transfer.

Reference [33] proposed an instance TL method based on K–L divergence measurements. They measured the similarity of the normal distribution between two domains and transformed this similarity into a transfer weight coefficient for the target subject.

Suppose that the normal distribution from the two datasets $N_{0}$ and $N_{1}$ can be expressed as:(1)$$N_{0}\sim N(\mu_{0},\Sigma_{0}),N_{1}\sim N(\mu_{1},\Sigma_{1})$$
where $\mu_{i}$ and $\Sigma_{i}$ are the mean value and variance (*i* = 1/0), respectively. The K–L divergence of the two distributions can be expressed as:(2)$$KL\left[N_{0}\right]\left[N_{1}\right]=0.5{\left[\left(\mu_{1}-\mu_{0}\right)\right]}^{T}\Sigma_{1}^{-1}\left(\mu_{1}-\mu_{0}\right)+trace\left(\Sigma_{1}^{-1}\Sigma_{0}\right)-ln\left(\frac{det\Sigma_{0}}{det\Sigma_{1}}\right)-K]$$
where *K* denotes the dimension of the data, μ represents the mean value, and Σ is the variance, *det* represents calculation of the determinant.

The similarity weight $\delta_{s}$ can be calculated by:(3)$$\delta_{s}=\frac{1/\bar{(KL}\left[N_{0},N_{1}\right]+\partial {)}^{4}}{\Sigma_{i=1}^{m}(1/{\left(\bar{KL}\left[N_{0},N_{1}\right]+\partial \right)}^{4})}$$
where ∂ is the balancing coefficient and KL is the summed divergence of the distribution characteristics of the target subjects. The results show that instance transfer can effectively reduce the calibration time and can significantly improve the average classification accuracy of MI tasks.

Li et al. [34] proposed importance-weighted linear discriminant analysis (IWLDA) with bootstrap aggregation. They defined the ratio r(x) of test and training input densities as transfer weight:(4)$$r\left(x\right)=\frac{P_{te}\left(x\right)}{P_{tr}\left(x\right)}$$
where $P_{tr}$ and $P_{te}$ represent the marginal probability distribution of the training set and the test set, respectively.

Then, they optimized the parameters of the LDA model by adding a regularization coefficient and transfer weights:(5)$$\min\overset{N}{\underset{i=1}{\sum }}r\left(x_{i}\right)(y_{i}-\hat{f}{\left(x_{i};\theta \right)}^{2})+\lambda \|\theta \|$$
where $y_{i}$ refers to the target labels corresponding to the feature vectors $x_{i}$ for *i*-th trials. Parameter θ is learned by least-squares.
(6)$$\min\overset{N}{\underset{i=1}{\sum }}{\left(y_{i}-\hat{f}\left(x_{i};\theta \right)\right)}^{2}$$
where
(7)$$X=\left(\begin{array}{c} \begin{array}{c} 1,x_{1}\\ 2,x_{2} \end{array}\\ .\\ .\\ .\\ .\\ n,x_{n} \end{array}\right)$$

The least-squares solution can be obtained by:(8)$${\hat{\theta }}_{IWLDA}={\left(X^{T}DX+\lambda I\right)}^{-1}X^{T}Dy$$
where *λ* (≥0) is the regularization parameter, *D* is the diagonal matrix with the *i*-th diagonal element, *I* is the identity matrix and ${\hat{\theta }}_{IWLDA}$ is the least-squares solution. They also combined the bagging method that independently constructs accurate and diverse base learners to improve the classification accuracy and to reduce the variance. The weighted parameters of the LDA model in the target domain can thus be optimized.

Covariate shift [35] is a common phenomenon in EEG processing across subjects/sessions. It is defined as follows: Given an input space *X* and an output space *Y*, the marginal distribution of $D_{s}$ is inconsistent with $D_{T}$, i.e., $P_{S}\left(x\right)\neq P_{T}\left(x\right)$. However, the conditional distribution of the two domains is the same, $P_{S}\left(y/x\right)=P_{T}\left(y/x\right)$. Covariate shift obviously affects the unbiasedness of a model in standard model selection, which reduces the generalization ability of the machine model during EEG decoding [30].

To address this issue, research has proposed covariate shift adaptation. For example, Raza et al. [36] proposed a transductive learning model based on the *k*-nearest neighbor principle. They initialized the classifier using data from the calibration stage and trained the optimal classification boundary. Then, adaptation was executed to update the classifier. The updated rules are as follows:

First, the Euclidean distance is used to measure unlabeled and labeled data:(9)$$dist_{\left(p,q\right)}=\sqrt{\sum_{j=1}^{m}{\left(q_{j}-p_{j}\right)}^{2}}$$
where *p* and *q* refer to the unlabeled and labeled data points, respectively, and *dist* is the Euclidean distance. Then, the *k*-nearest neighbors are selected based on the Euclidean distance. Next, this distance is converted to inverse form $dist_{inv\left(i\right)}$, which represents the corresponding pattern in the training database that is closer to the current unlabeled feature set.
(10)$$dist_{inv\left(i\right)}=\frac{1}{d_{\left(q,p\right)}^{i}+\epsilon }$$
where *i* is the label and ϵ=0.001 is the bias. To decide if the current trial’s features and estimated label should be added to the existing knowledge base, a confidence ratio *CR* is calculated:(11)$$CR_{j}=\frac{\sum_{1}^{k}dist_{inv\left(i\right)}\left(l\left(i\right)==j\right)}{\sum_{1}^{k}dist_{inv\left(i\right)}}$$

The *CR* index is calculated to predict the label for the unlabeled test data. The predicted test data are then added into the knowledge database, following which the decision boundary is recalculated to realize the update.

#### 3.4.2. Transfer Learning Based on Feature Representation

TL based on feature representation can be achieved by reducing the difference between two domains by feature transformation or projecting the feature from two domains into the uniform feature space [37,38,39]. Unlike instance transfer, feature representation TL aims to encode the shared information across subjects/sessions into a feature representation. For example, spatial filtering and time–frequency transformation are used to transform the raw data into feature representations.

Nakanish et al. proposed a spatial filtering approach called the task-related component analysis (TRCA) method to enhance the reproducibility during SSVEP tasks and to improve the performance of an SSVEP-based BCI [40].

Suppose that two domain signals consist of two parts: A task-related signal s(t) and a task-unrelated signal z(t). A multichannel signal from x(t) can be calculated as:(12)$$x_{i}\left(t\right)=a_{1,i}\;s\left(t\right)+a_{2,i}z\left(t\right),\;i=1,\;2,\;3\ldots n$$
where *i* represents the number of channels and *a* refers to the project coefficients; 1 and 2 represent labels.
(13)$$y\left(t\right)=x_{i}\left(t\right)\overset{n}{\underset{i=1}{\sum }}x\left(t\right)=\overset{n}{\underset{i=1}{\sum }}(a_{1,i}\;s\left(t\right)+a_{2,i}z\left(t\right))$$
where y(t) refers to the target data, and the optimization goal is to solve $a_{1,i}=1$ and $a_{2,i}=0$. The covariance between the $j_{1}-th$ and the $j_{2}-th$ trials is described as:(14)$$c_{j_{1},j_{2}}=Cov\left(y^{\left(j_{1}\right)}\left(t\right),y^{\left(j_{2}\right)}\left(t\right)\right)=\overset{n}{\underset{i_{1}i_{2}=1}{\sum }}w_{i_{1}}w_{i_{2}}Cov\left(x^{\left(j_{1}\right)}\left(t\right),x^{\left(j_{2}\right)}\left(t\right)\right)$$

All combinations of the trials are summed as:(15)$$\overset{N_{t}}{\underset{j_{1},j_{2}=1,j_{1}\neq j_{2}}{\sum }}c_{j_{1},j_{2}}=\omega^{T}S\omega$$
where *j* represents the number of trials and ω refers to the spatial filters. Matrix *s* is defined as:(16)$$s_{i_{1},i_{2}}=\overset{N_{t}}{\underset{i_{1},i_{2}=1,i_{1}\neq i_{2}}{\sum }}Cov\left(x_{i_{1}}^{j_{1}}\left(t\right),x_{i_{2}}^{j_{2}}\left(t\right)\right)$$

The variance of y(t) is constrained to obtain a finite solution:(17)$$\mathrm{Var}\left(\sum \left(t\right)\right)=\omega^{T}Q\omega =1$$

The optimization is calculated as:(18)$$\tilde{\omega }=\mathrm{argmax}\;\frac{\omega^{T}S\omega }{\omega^{T}Q\omega }$$
where $\tilde{\omega }$ is the optimal spatial filter. Finally, the correlation coefficient is calculated by Pearson’s correlation analysis between the data from the two domains. In their study, spatial filters as a feature representation were transferred to the target domain. The results showed that this method significantly improves the information transfer rates and classification accuracy. Based on this research, Tanaka [41] improved the TRCA method by maximizing the similarity across group of subjects, and they named this novel method group TRCA. The results showed that the group representation calculated by the group TRCA method achieve high consistency between two domains and offer effective data supplementation during brain–computer interaction.

CSP is a popular method for feature extraction, which is often used for MI classification. During calculation, a spatial filter is adopted to maximize the separation between the class variances of EEG. However, heterogeneous data across subjects/sessions causes poor classification performance of the model in the training stage. One feasible approach to solve the limitation is regularization. Lotte [42] presented regularized CSP to improve the classification accuracy across subjects. In their study, they discussed two strategies. One of them was regularizing the covariance matrix estimated. They can be, respectively, expressed as:(19)$${\tilde{S}}_{i}=\left(1-\gamma \right){\tilde{S}}_{i}+\gamma I$$
(20)$${\hat{S}}_{i}=\left(1-\beta \right)c_{i}S_{i}+\beta D_{i}$$
where $S_{i}$ represents the initial spatial covariance matrix for class *i*, ${\tilde{S}}_{i}$ is the regularized estimate, *I* is the identity matrix, $c_{i}$ is a constant scaling parameter, and $D_{i}$ represents the generic covariance matrix. The regularization parameters can be defined as γ and β. This strategy aims to optimize the covariance matrix by transforming other subjects’ data into covariance combined with the regularization parameters and by transferring this feature to the target subject.

Another approach is regularizing the CSP objective function. CSP uses spatial filters ω to extremize the function:(21)$$J\left(\omega \right)=\frac{\omega^{T}C_{1}\omega }{\omega^{T}C_{2}\omega }$$
where $C_{i}$ is spatial covariance matrix from class *i*. This approach optimizes CSP algorithms by regularizing the CSP objective function itself:(22)$$G_{P_{1}}\left(\omega \right)=\frac{\omega^{T}S_{1}\omega }{\omega^{T}S_{2}\omega +\partial P\left(\omega \right)}$$
where P(ω) represents a penalty function for the measurement distance between the spatial filter and the prior information. The goal of the objective function is to maximize $G_{P_{1}}\left(\omega \right)$ and to minimize P(ω). ∂ is a user-defined regularization parameter. The prior information from the source domain provides a good solution to guide the optimization direction of the estimation of spatial filters.

In addition, adaptation regularization is a typical feature TL method based on the structural risk minimization principle and the regularization theory. Cross-domain feature transfer is mainly operated by three methods: (1) Utilize the structural risk minimization principle and minimize the structural risk functional; (2) minimize the distribution difference between the joint probability distributions; (3) maximize the manifold consistency underlying the marginal distributions [43]. In recent research, Chen et al. [44] developed an efficient cross-subject TL framework for driving status detection. They used adaptation regularization to measure and reduce the difference of the features from the two domains and to extract the features by filtering algorithms. The results showed that this framework can achieve high recognition accuracy and good transfer ability.

#### 3.4.3. Transfer Learning Based on Model Parameters

The assumption of model parameter TL is that individual models across subjects/sessions should share some parameters. The key step of this approach is to find shared parameter information and to realize knowledge transfer. The domain adaption (DA) of a classifier is the common method of model parameter transfer. The knowledge of the parameter information from $D_{s}$ is reused and adjusted according to the prior distribution of $D_{T}$ [45]. A DA method, named adaptive extreme learning machine (ELM), was proposed by Bamdadian et al. [46]. ELM is a single-hidden layer feedforward neural network, which determines the output weights by operating the inverse operation of the hidden layer weight matrices [47]. This method has two steps: First, the classifier is initialized by data from the calibration session. Then, the update rule for the output weight based on least-square minimization is calculated. The update rule is calculated as follows:

The initial output weight α can be defined as:(23)$$\alpha =H^{+}T=\varphi^{-1}H^{T}T$$
where *H* is the output matrix of hidden layer, $\varphi =H^{T}H$ and $H^{+}$ refer to the Moore–Penrose pseudo-inverse of *H*, and *T* represents the label category. The updated weight $\alpha^{m+1}$ is calculated as:(24)$$\alpha^{m+1}=\alpha^{m}+\varphi_{k+1}^{-1}H_{k+1}^{T}\left(T_{k+1}-H_{k+1}\alpha^{m}\right)$$
(25)$$\varphi_{k+1}=\varphi_{k}+H_{k+1}^{T}H_{k+1}$$
(26)$$\varphi_{k+1}^{-1}=\varphi_{k}^{-1}-\varphi_{k}^{-1}H_{k+1}^{T}\left[I+H_{k+1}\varphi_{k}^{-1}H_{k+1}^{T}\right]H_{k+1}\varphi_{k}^{-1}$$
where *k* is *k*-th hidden node, φ is orthogonal matrix calculated by H. The experiential results showed that adaptive ELM can significantly improve the classification accuracy in MI classification across subjects.

Another strategy is ensemble learning, which combines multiple weak classifiers from the source domain into a strong classifier. Dalhoumi et al. [48] proposed a novel ensemble strategy based on Bayesian model averaging. They calculated the probability of having a class label $y_{q+1}$ given a feature vector $h_{q+1}$:(27)P(yq+1/xq+1)=∑n=1NP(yxq+1n,jn)P(jnT)
where $x_{q+1}^{n}$ is the logarithmic variance feature vector, $j_{n}$ is a set of hypotheses from the source domain, and *T* is the test set. The hypothesis prior P(jnT) is estimated in the following method:(28)$$w^{*}=argmin\overset{p}{\underset{p=1}{\sum }}l\left(\overset{N}{\underset{n=1}{\sum }}j^{n}\left(x_{p}^{n}\right)y_{p}\right)$$
(29)p(jnT)=wn*
where $x_{p}^{k}$ is the projection of the feature vector *x* on the spatial filters of subject *k*. The learned ensemble classifier can be used to predict labels for the target user:(30)$$h^{*}=\overset{N}{\underset{n=1}{\sum }}w_{n}^{*}j_{n}$$

The results showed that this ensemble strategy can improve the classification performance in small-scale EEG data by evaluation on a real dataset.

In recent years, deep neural networks have provided good results for the processing of EEG signals [49,50]. Due to their end-to-end model structure and automatic feature extraction ability, deep neural networks minimize the interference of redundant information and improve the classification performance. Inspired by computer vision, a deep neural network learns generic feature representations by lower layers of the model. Specific feature representations with the relevant specific subjects or sessions are learned by the high layer [51]. Therefore, freezing lower layers and fine-tuning higher layers is a good way to realize model parameter transfer based on deep learning.

Zhao et al. [52] proposed an end-to-end deep convolution network for MI classification. To avoid the limitation of a small sample and overfitting, they utilized the data from $D_{s}$ to pre-train the source network and to transfer the parameters of several layers to initialize the target network. First, the network was pre-trained using data from the source domain. Then, they used the *M* source subjects $W^{s}$ to initialize the *n*th layer’s target network by a weight average:(31)$$W_{n}^{t}=\overset{M}{\underset{m=1}{\sum }}\rho_{m}W_{mn}^{s}$$
where ρ represents the strength of the source network and $W_{mn}^{s}$ refers to the connecting weights of the *n*th layer to the next layer. The next stage is to fine-tune the target initialized network by data from $D_{T}$. The results showed that the parameter transfer strategy can reduce the calibration time for new subjects and can help the deep convolution network to obtain better classification performance.

Raghu et al. used CNN combined with TL to recognize epileptic seizures [53]. They proposed two different transfer methods: To finetune a pre-trained network and then extract image features by said pre-trained network, and to classify the status of brain using an SVM. Popular networks such as Alexnet, VGG16net, VGG19net, and Squeezenet, were used to verify the performance of the proposed framework.

The summary of collected studies is shown in Table 3.

## 4. Discussion

Based on the numerous papers surveyed herein, we briefly summarized the development of the application of TL to EEG decoding. This will help researchers scan the status of this field and receive useful guidance in future work.

According to the various studies surveyed in this paper, it is not hard to determine the points of interest that researchers focus on. As shown in Figure 2, more studies have focused on active BCI (i.e., MI, SSVEP, and ERP) among these different EEG paradigms. One possible explanation is that the goal of these mental activity decoding studies is to categorize EEG from different classes. This would allow many machine learning methods to be applied to this paradigm. From Table 2, it can be seen that the application scenarios of TL in the existing literature have focused almost only on classification and regression tasks.

The method of model parameter transfer is not applicable to only a few subjects with initially low BCI performance. The feature of EEG from these subjects exhibits inseparability in feature space. Therefore, the parameter optimization of the classifier does not significantly improve the classification results. It is worth noting that the adaptive strategy of the classifier should be considered a supplement to achieve the goal of a calibration-free mode of operation [123]. The combination of TL and the adaptive strategy may receive increasing attention in future studies.

It is also worth noting that TL showed good results across subjects/experiments, but the detail of variability across sessions/subjects was unclear. Some studies proposed that the Bayesian model is a promising approach to capture variability. This model is built based on multitask learning, and variation in some features is often extracted, such as spectral and spatial [124,125].

Due to its end-to-end structure and competitive performance, deep learning has been successful in processing EEG data [126]. However, the computational power and small-scale data are a limitation during practical operation. A hybrid structure based on TL and deep learning is a promising way to address this issue. For example, one of the methods is fine-tuning the pre-trained network, which has proven to be effective. With the development of deep learning technology, the research for such a hybrid structure is still a hot topic for future research.

As reported in the above-cited studies, TL is instrumental in EEG decoding across subjects/sessions. However, knowledge transfer across tasks/device is still a blank field. This issue is worth exploring and will make EEG-based BCI systems much more practical.

## 5. Conclusions

In this paper, we reviewed the research on TL for EEG decoding that was published between 2010 and 2020. We discussed numerous approaches that can be divided into three categories: Instance transfer, feature representation transfer, and parameter of classifier transfer. Based on the summary of their results, we can conclude that TL can effectively improve the decoding performance in classification and regression tasks. In addition, TL provides adequate performance in initializing BCI systems for a new subject, which reduces the length of time of the calibration process. Although there are some limitations for using TL for EEG decoding, such as the scope of application of TL and suboptimal performance on some occasions, TL shows strong robustness. Overall, TL is instrumental in EEG decoding across subjects/sessions. In addition, achieving a calibration-free model of operation and higher accuracy of decoding are worthy of further research.

## Figures and Tables

**Figure 1 sensors-20-06321-f001:**
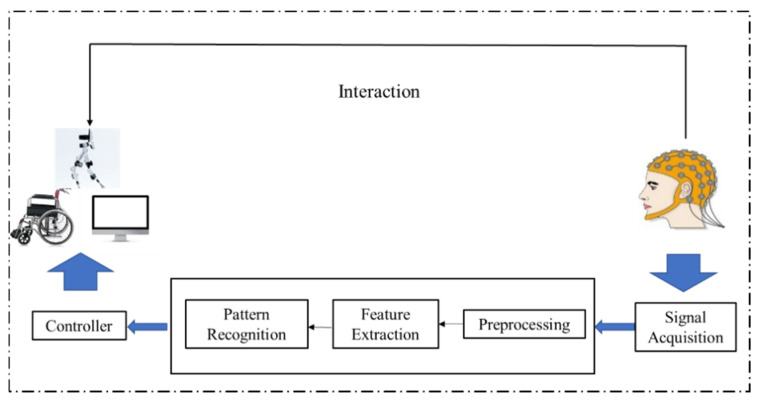
Framework of an electroencephalography (EEG)-based brain–computer interface (BCI) system.

**Figure 2 sensors-20-06321-f002:**
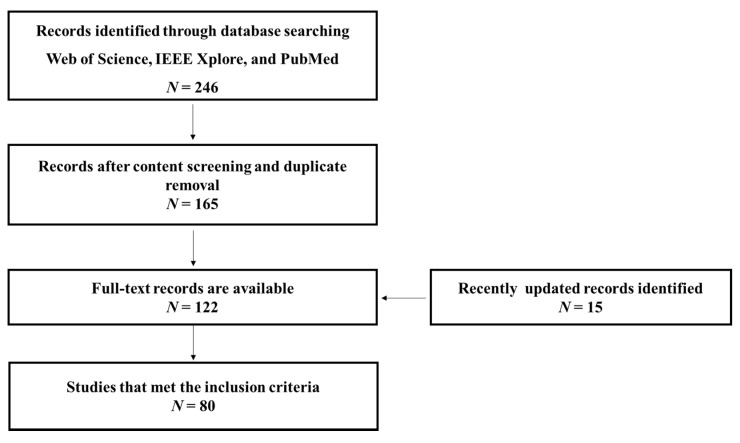
The search method for identifying relevant studies.

**Figure 3 sensors-20-06321-f003:**
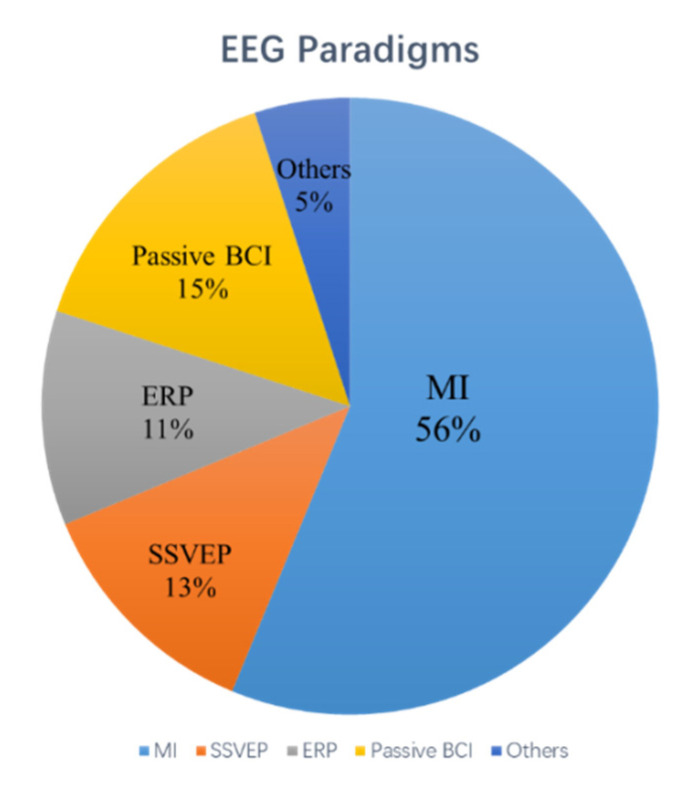
The percentage of different EEG pattern strategies across collected studies.

**Figure 4 sensors-20-06321-f004:**
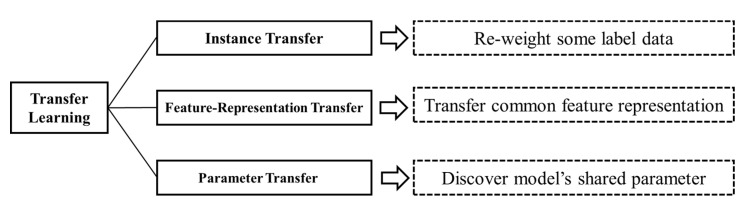
Different approaches to transfer learning.

**Table 1 sensors-20-06321-t001:** Inclusion and exclusion criteria.

Inclusion Criteria	Exclusion Criteria
Published within the last 10 years (as transfer learning (TL) for EEG has been proposed and developed in recent years).	A focus on the processing of invasive EEG, electrocorticography (ECoG), magnetoencephalography (MEG), source imaging, fMRI, and so on, or joint studies with EEG.
A focus on non-invasive EEG signals (as the object for discussion in this review).	No specific description of TL for EEG processing.
A specific explanation of how to apply TL to EEG signal processing.	\

**Table 2 sensors-20-06321-t002:** Dataset.

Datasets	Task	Subject	Channel	Amount of Data (Per Subject)	Sampling Rate	Reference
BCIC-II-IV	2 MI classes	1	28	3 sessions/416 trials	1000 Hz	[19]
BCIC-III-II	P300	2	64	5 sessions	240 Hz	[20]
BCIC-III-IVa	2 MI classes	5	118	4 sessions/280 trials	1000 Hz	[21]
BCIC-IV-2a	MI	9	25	2 sessions/288 trials	250 Hz	[22]
BCIC-IV-2b	2 MI classes	9	6	720 trials	250 Hz	[23]
P300 speller	P300	8	8	5 sessions/20 trials	256 Hz	[24]
DEAP	ER	32	40	125 trials	128 Hz	[25]
BCIC -III- IVC	MI	1	118	630 trials	200 Hz	[26]
SEED	ER	15	64	3 sessions/15 trials	200 Hz	[27]
OpenMIIR	Music Imagery	10	64	5 sessions/12 trials in four tasks	512 Hz	[28]
CHB-MIT	ED	22	23	844 hours’ collection	256 Hz	[29]

**Table 3 sensors-20-06321-t003:** Summary of transfer learning for EEG decoding.

Pattern	Reference	Type	Transfer Method	Feature Extraction	Datasets	Results
MI	[33]	ITL	Similarity measurement with KL divergence	LR+CSP	19 subjects’ data, BCIC-IV-2a, and BCIC-III-4a	70.3% and 75% and 75%
MI	[54]	ITL	Informative subspace transferring and selective ITL with active learning	LDA	BCIC-IV-2b	\
MI	[55]	FTL	Ensemble learning and adaptive learning	LDA	NIPS	68.1%
MI	[56]	ITL	Similarity measurement with Jensen Shannon ratio and rule adaptation TL	CSP+LDA	BCIC-IV-2a	77%
MRP	[57]	ITL	MMD and regularized discriminative spatial pattern	Linear RR	BCIC-I-1 and BCIC-II-4	\
P300	[58]	ITL	Ensemble learning generic information	Bayesian LDA	8 participants’ data	62.5%
SSVEP	[59]	ITL	Variability assessment Fisher’s discriminant ratios	Cluster	8 subjects’ data	\
P300	[60]	ITL	Dynamically adjusts weights of instances	Liner SVM	BCIC-III-2 and dataset of P300 speller	74.9%
MI	[61]	ITL	Selective informative with normalized entropy	LDA	BCIC-IV-2b	75.8%
MI	[62]	ITL	Selective informative expected decision boundary	LDA	BCIC-IV-2a	75.6%
MI	[63]	MTL	Domain adaptation parallel transport on the cone manifold of SPD	Linear SVM	BCIC-IV-2a	\
DD	[64]	ITL	Selective transfer learning	Linear regression	15 subjects’ data	66%
MI	[65]	ITL	Composite local temporal correlation CSP Frobenius distance	Liner quadratic Mahalanobis	BCIC-III-IVa	89.21%
SSVEP	[66]	ITL	Ensemble learning and similarity measurement with mutual information	LDA	10 healthy subjects’ data	\
Cognitive detection	[67]	ITL	Similarity measurement by Pearson’s correlation coefficient	SVM	28 subjects’ data	87.6%
ERP	[68]	FTL	Probabilistic zero framework	Unsupervised adaptation	Akimpech dataset	\
MI	[69]	FTL	DA with power spectral density	CNN	BCIC-IV-2a	\
MI	[70]	FTL	Many-objective optimization	Linear SVM	BCIC-III-IVa	75.8%
VEP	[71]	FTL	Active semi-supervised TL	SVM	14 subjects’ experiments	\
MI	[72]	FTL	Adaptive Selective CSP	Discriminant analysis	6 participant experiments	61%
MI	[34]	FTL	CSA	Importance-weighted LDA	BCIC-III dataset	79.1%
MI	[73]	ITL	Instance TL based p-hash	CNN	BCIC-IV-2b	
MI	[74]	FTL	Informative TL with AL	LDA	BCIC-IV-2a	67.7%
MI	[75]	MTL	Modifications of CSP	SVM	BCIC-III-IVa	\
MI	[76]	MTL	*k*-nearest neighbors principle	SVM+LDA	BCIC-IV-b	\
ER	[77]	FTL	Transfer recursive feature elimination	Least-squares SVM	DEAP dataset	78%
SSVEP	[78]	MTL	Least-squares transformation	\	8 participant experiments	82.1%
MI	[79]	FTL	Domain transfer multiple kernel boosting	SVM	BCIC-III-Iva 5 subjects’ data	81.6% 76%
SSVEP	[80]	FTL	Spatial filtering transfer	TRCA	10 subjects’ data	\
SSVEP	[81]	FTL	Reference template transfer	MestCCA	10 subjects’ data	\
SSVEP	[82]	FTL	Reference template transfer	Transfer template-CCA	12 subjects’ data	85%
SSVEP	[83]	FTL	Reference template transfer	Adaptive combined-CCA	10 subjects’ data	83%
MI	[84]	FTL	Fuzzy TL based on generalized hidden-mapping RR	SVM	BCIC-IV-2a	89.3%
MI	[46]	MTL	Adaptive extreme learning machine	SVM+ELM	12 subjects’ data	71.8%
MI	[48]	MTL	Classifier ensemble	LDA	BCIC-IV-2a	\
MI	[13]	FTL	Regularized CSP with TL	LDA	BCIC-III-IVa	78.9%
Multitask	[85]	FTL	Geometrical transformations on Riemannian Procrustes analysis	/	8 publicly available BCI datasets	\
ERP	[86]	FTL	Spectral transfer using information geometry	MDRM	15 subjects’ data	62%
MI	[87]	FTL	Space adaptation	LDA	BCIC-IV-2a	77.5%
MI	[88]	FTL	Feature space transformation	LDA	PhysioNet datasets	72%
MI	[89]	FTL	Tangent space-based TL	LDA	BCIC-IV-2a	\
ER	[90]	FTL	Transfer component analysis and kernel principle component analysis	SVM	SEED	77.96%
MI	[91]	FTL	Transfer kernel CSP	SVM	BCIC-III-IVa	81.14%
MI/ ERP	[92]	FTL	Affine transform	Minimum distance mean and Bayesian classifier	BCIC-IV-2a and Brain Invaders experiment	\
SSVEP	[93]	ITL	Riemannian similarities	Bootstrapping	12 subjects’ data	80.9% & 81.3%
MI	[94]	FTL	Multitask learning	RR+SVM	10 healthy subjects’ data and an ALS subject’s data	85%
Imagined speech	[95]	ITL	Inductive transfer learning	Naïve Bayesian classifier	27 subjects’ data	68.9%
MI	[96]	FTL	Transferable discriminative dimensionality reduction	KNN+SVM	5 subjects’ data	74.4%
MI	[7]	FTL	Nonstationary information transfers	LDA	5 subjects’ data and BCIC-III-IVa	80.4%
MI	[97]	MTL	Fine-tuned based on VGG16	CNN	BCIC-IV-2b	74.2%
MI	[98]	MTL	Fine-tuned based on pre-trained network	CNN	BCIC-IV-2a	69.71%
ErrPs	[99]	MTL	Fine-tuned based on pre-trained network	CNN	15 epilepsy patients’ data	81.50%
Music Imagination	[100]	MTL	Fine-tuned based on AlexNet	LSTM	OpenMIIR dataset	30.83%
ErrPs	[101]	MTL	Fine-tuned based on pre-trained network	CNN	31 subjects’ data	84.1%
DD	[102]	ITL	Source domain selection	Weighted adaptation regularization	16 subjects ‘data	\
Attention detection	[103]	ITL	Subject adaptation	CNN	8 subjects ‘data	84.17%
MI	[52]	ITL	Subject transfer	CNN	BCIC-IV2a and BCIC-IV-2b	0.56 and 0.65 (MK)
MI	[104]	MTL	Fine-tuned based on pre-trained network	RBM	BCIC-IV2a and 12 subjects’ data	88.9%
P300	[105]	MTL	Fine-tuned based on pre-trained network	CNN	BCIC-III-2	90.5%
MI	[106]	MTL	Fine-tuned based on pre-trained network	CNN	BCIC-IV-2b	0.57 (MK)
MI	[107]	MTL	Fine-tuned based on pre-trained network	Conditional variational autoencoder	PhysioNet datasets	73%
Music Imagination	[108]	FTL	Cross-domain encoder	Attention decoder-RNN	OpenMIIR datasets	37.9%
MI	[36]	ITL	Covariate shift detection and adaptation	Linear SVM	BCIC-IV-2a BCIC-IV-2b	73.8% and 69.7%
MWA	[109]	FTL	Cross-subject statistical shift	Random forest	9 subjects‘ data	\
MI	[110]	MTL	Fine-tuned based on multiple network	CNN	BCIC-IV-2a and BCIC-IV-2b	\
MI	[111]	MTL	Four-strategy model transfer learning	Deep neural network	BCIC-IV-2a	\
MI	[112]	ITL	Subject–subject transfer	CNN	3 subjects’ data	\
P300	[113]	ITL	Subject–subject transfer	Linear SVM	22 subjects’ data	68.7%
ER	[114]	ITL	Measurement on Riemannian geometry instance transfer	SVM	MDME and SDMN datasets	\
DD	[40]	FTL	Adaptation regularization based TL	Multiple classifier	23 subjects’ data and NIH physio bank	89.59%
ED	[53]	MTL	Fine-tuned based on pre-trained network	GoogLeNet and Inception v3	TUH open-source database	82.85% and 88.30%
ED	[115]	MTL	Fine-tuned based on pre-trained network	CNN and bidirectional LSTM	CHB-MIT EEG dataset	99.6%
ED	[116]	MTL	Fine-tuned based on pre-trained network	CNN	CHB-MIT EEG dataset	92.7%
MI	[117]	FTL	Spatial filtering transfer and matrix decomposition	ELM	BCIC-III-IVa and BCIC-IV-1	89% 62%
SSVEP	[41]	FTL	Spatial filtering transfer	Group TRCA	Benchmark dataset	\
MI	[118]	MTL	Adversarial inference	CNN	52 subjects’ data	\
ER	[119]	FTL	Power spectral density feature	Polynomial/Gaussian kernels/ naïve Bayesian SVM	DEAP, MAHNOB-HCI, and DREAMER	\
MWA	[120]	FTL	Ensemble learning	Stacked denoising autoencoder	8 subjects’ data	92%
MI	[121]	FTL	Data mapping and ensemble learning	LDA	BCIC-IV-2a	0.58 (MK)
MI	[122]	FTL	Center-based discriminative feature learning	CNN	BCIC-III-IVa	76%

“\” represents that there is no specific description or else multiple descriptions for the results.

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
