# Peer review of "Application of Transfer Learning in EEG Decoding Based on Brain-Computer Interfaces: A Review"

_sensors, 2020, doi:10.3390/s20216321_

Round 1

Reviewer 1 Report

The manuscript presents a review of the use of transfer learning for electroencephalography (EEG) decoding. Being a “review” of a current topic, I believe the work has merit. However, due to the current structure of the manuscript, it is likely that a significant improvement needs to be made before the paper is fit for publication. I believe the authors are on the right track for a good piece of work, but it currently seems lacking.

Authors say in lines 42-46: “Research [2] show that the brain-generated potentials recorded on the scalp associated with specific task conditions are represent by a 2D (time and channel) matrix. While machine learning (ML) shows its powerful performance in matrix computing field. Thus, machine learning has been applied on decoding for EEG signals due to its highly structure.”. I suggest reviewing for style (“Research [2]” seems a vague reference) and would also suggest more attention here. Many data can be translated into matrices or tensor, like graphs, 3d data, etc. Machine learning is used in many different fields of science that, not necessarily, rely on 2d matrices-tabular data.

The idea of Table 2 is very interesting, but I believe the authors need to add some more thought on the information that will be provided. For example, is “416 epochs” related to the training process (as commonly used in machine learning literature) or to the dataset itself? I believe some polishing on this table (including the format) could improve the value this manuscript.

Equation 1, N_0 and N_1 are not defined. For other equations, P_te, P_tr are not defined. Omega_s is used once to define similarity weight, then omega(x) is used to define “the ratio of test and training input densities”. What are test and training densities? ∈ was used at least twice with different meanings. The whole 3.4.1 needs to be revised due to these inconsistencies. Section 3.4.2 uses omega again, now with a third meaning apparently. Section 3.4.2 also has missing variables definition. Same for section 3.4.3. It is unfortunate that such lack of consistency exists in these sections. This deters the objective of the manuscript to serve as a guide for other researchers working with transfer learning for EEG.

Overall, the paper needs polishing in style. I marked some suggestions below, but they are not all the occurrences. The authors need to define whether or not they will use TL for transfer learning. TL is used in the abstract, then again in section 3.4.2, but TL was not defined in the main text before section 3.4.2. Does “researcher” in table 3 mean “reference number”? The bibliography format seems to be inconsistent.

Minor comments:

Line 9: “The algorithms of electroencephalography (EEG) decoding is”: the algorithms… are

Lines 13-14: “result in difference in feature distribution”: please review

Lines 17-21: “A review of literature on TL application for EEG decoding was proposed to report what kind of TL methods were used (instance knowledge, feature representation knowledge and model parameter knowledge), describe which type of paradigms of EEG are analyzed, datasets for performance evaluation.”: please review, this does not seem to be a complete sentence although is long.  

Lines 35-36: “A typical non-invasive BCIs system framework based on EEG is shown in figure 1”: as a suggestion, try to maintain active voice as much as possible: Figure 1 shows a typical non-invasive BCIs system framework based on EEG. This is an “overall” suggestion for when possible.

Line 52: “But it is often violated in the flied of bioelectric signal processing.”: typo (flied). Please review for style, “it” here might not be very clear for non-native English speaker.

Lines 53- 54: “Because differences in physiological structure and psychological states may cause obvious variation of EEG.” Please review. This sounds like an incomplete sentence.

Line 61: “However, large and high-quality datasets is”: datasets are

Line 63: “One promising approach to solve these problems is transfer learning”: first occurrence of ‘transfer learning’ in the main text – TL should be defined here again. Or remove TL from the abstract.

Line 67: “and learning task ?_s, a target domain ?_T and learning task ?_T. Transfer” please review. Remove the period before transfer or the previous sentence is incomplete.

Lines 79-83: “ What problems does transfer learning solve for EEG decoding? (Section 3.1) Which paradigms of EEG are used to analysis for transfer learning? (Section 3.2) which datasets can we refer to verify the performance of these methods? (Section 3.3). What types of transfer learning frameworks are available? (Section 3.4)” please review for consistency. Are you going to use a period after section parentheses ‘(Section 3.3).’ or not? One of the ‘which’ is not capitalized.

Table 1: “This review focuses on noninvasive EEG signals collected from the human.” Please review, incomplete sentence.

Line 107: “Although advanced algorithms such as machine learning and deep learning”: I would deep learning is part of machine learning and that machine learning is a methodology or technique rather than an algorithm. Please review.

Lines 108-109: “it still suffers from some limitations that hinder its widely application to the practice.” What does ‘it’ refer to here?

Lines 109-111: “One of major hypotheses in machine learning is that training data and test data belong to the same feature space and follow the same probability distribution.” Repetition of abstract and introduction. I understand this sentence is here to be connected with the following one (“In the field of biomedical…”) but I suggest the sentence in lines 109-111 be revised for style so that it is not so similar to the ones written before.

Line 179: “Reference [4] proposed”: I think this would read better as “Pan et al. [4] proposed”. I think you might need to review your references as it does not seem they are following any standard.

Line 197: “two domains and transform the similarity into transform weight for target subject”: is “transform weight” what you really meant? Please review for clarity.

Lines 230-231: “EEG decoding. [Sugiyama et al., 2007].” Inconsistent referencing style.

Line 255: “feature space. [34, 35, 36].” Remove period before reference.

Line 255: “Differ from instance transfer”: do you mean “unlike instance transfer”?

Line 385: “SVM.” Support vector machine was not defined in the main text.

Line 392: “we have brief”: we have briefly

Line 394: “help researcher”: help researchers

Reviewer 2 Report

This is interesting paper which reviews all the work that has been done on the analysis of EEG signal using the latest deep learning algorithms and transfer learning. However, the scope is very limited.

The authors have reviewed many papers. The authors have addressed motor imagery, evoked potential, event related potential, and passive BCI. However, other EEG analysis have also been used for assessing anaesthesia, apnea, brain death etc.

Since this is a survey paper, such topics should not be ignored. In addition, the paper is written in a way that is difficult to understand. The English need to be revised and rewritten by a professional person. The journal does provide English rephrasing service, which I think it is worth using to improve the language.

Reviewer 3 Report

The author made a comprehensive literature review of the BCI researches in the past ten years. From the perspective of transfer learning, the algorithms related to EEG research are divided into three categories. Each algorithm of each type of transfer learning is summarized. The authors also made a summary with a table at the end. This paper is very detailed and covers almost all commonly used algorithms for EEG classification or recognition problems. Thus, this paper is very helpful for BCI researchers. To improve the quality, I'd suggest the author add one more section about the future direction regarding the transfer learning to EEG applications.

Round 2

Reviewer 1 Report

I would like to thank the authors for their effort to address my comments from the first review. 

There is an improvement in table 2. I still believe it could be more polished though. For example, the fields "subject", "sessions", "sampling rate" seem to be present for all (or most) rows. Wouldn't be more adequate to transform those into columns and make it easier for the reader to reference?

I strongly suggest the authors add a table for symbols for all the equations, something like what nicely done for the acronyms in Appendix: List of acronyms. It seems some variables are still not defined, although this is a significant improvement than the first version. 

In general I suggest the authors perform a review to polish the manuscript as they see fit. I made some notes below, but they are not extensive. 

Minor comments:

Lines 76-77: There are two main scenarios in EEG-based BCIs, namely, cross-subject transfer and cross-session transfer. The goal of this method is to find the similarity...": what does "this" refer to considering the previous sentence talks about two main scenarios?

Lines 80-81: "In this study, we attempted to summarize the transferred knowledge..." then lines 84-85: "This review of TL applications for EEG classification attempted to address the following critical questions:..." please define the objectives more clearly. 

Equation (9): * is the symbol for convolution. Is that what you mean?

Equations (1) vs (15): two different symbols for variance?

Equation (16): need to define omega tilde

Is phi defined for equations 23, 24?

Reviewer 2 Report

The correction have improved the paper and made the presentation clearer. 

The author have used active voice, please change the style to passive voice. 
